# Evaluating the Cost of Employing LPs and STPs in Planning: Lessons Learned From Large Real-Life Domains

**Elad Denenberg** and **Amanda Coles** and **Derek Long**

Department of Informatics, King's College London, UK.

email: {elad.denenberg,amanda.coles,derek.long}@kcl.ac.uk

## Abstract

When solving real-life problems we often encounter issues that are not captured by academic benchmark domains. In this paper we consider an application problem, representative of a class of real-world problems that have interesting properties: long solution plans with many temporal/numeric constraints. We identify a number of limitations of a popular family of planners in solving these problems. This family of planners perform Forward Search and call a Linear Programming (LP) solver multiple times at every state to check for consistency, and to set bounds on the numeric variables in order to determine action applicability. These checks during search allow the pruning of branches; however, they do carry computational cost. In this paper we investigate and analyse this trade-off, with particular reference to our class of application problems, and show that adapting the planners to call the LP solver less often, and using a cheaper consistency check at each state, can improve performance.

## 1 Introduction

Automated Planning is concerned with using a planner to formulate a sequence of actions that transforms a given initial state into a desired goal state. One strength of planning is domain-independence: a single general planner can plan in a wide range of different application domains. For example, space (Chien et al. 2000), battery usage (Fox, Long, and Magazzeni 2011) and software penetration testing (Obes, Sarraute, and Richarte 2013). In order to facilitate their application in realistic problems planners need to reason with expressive models of the world. Such models can be temporal: finding a plan with timestamped actions, taking into account action durations and concurrency; as well as numeric: considering numeric variables that change discretely, or in hybrid problems also continuously, over time (Fox and Long 2003).

When planning in expressive domains a planner needs a mechanism to schedule the plan, i.e. assign timestamps to actions to meet the temporal and numeric constraints. One option is the decision epoch mechanism of SAPA (Do and Kambhampati 2001) and Temporal Fast Downward (Eyerich, Mattmüller, and Röger 2009); alternatively, one can use simple temporal networks (STNs) (Dechter, Meiri, and Pearl 1991) e.g. as in the planner Crikey 3 (Coles et al. 2009). Approaches to hybrid planning employ more complex mechanisms include compiling the problem into SAT Modulo Theories (Cashmore

et al. 2016), interval relaxation (Scala et al. 2016), time discretization (Piotrowski et al. 2016) and convex optimization (Fernández-González, Karpas, and Williams 2017).

In this work we focus on a family of planners that make use of LP solvers to schedule the plan. This family includes COLIN (Coles et al. 2012), POPF (Coles et al. 2010) and OPTIC (Benton, Coles, and Coles 2012). These planners perform forward state-space search starting from the initial state. At each state in the search a LP solver is used once to determine whether there exists a consistent schedule for the plan (adhereing to temporal and numeric constraints). If a no consistent schedule exists for the plan to the current state, the search branch can be pruned. If the state is consistent, the LP is then used multiple more times to bound the numeric variables and thus prune the space of applicable actions in this state, narrowing the search space ahead.

In this work, we analyse the performance of these planners on a particular interesting class of real-world application problems, illustrated in a domain provided by our industrial partner. This domain highlights issues they encountered in using this family of planners in their deployed applications. Specifically, the problems require long solution plans, with large numbers of numeric variables and temporal constraints. An important observation made in this class of application problems is that whilst the traditional planning benchmarks, on which the planners were initially evaluated, lead typically to small LPs being generated at each state (Coles et al. 2012); LP solving in this class of application domains is much more expensive as the long plans, hence large numbers of variables, lead to larger LPs, which hinders planner performance.

In this paper we consider the trade-offs involved in solving LPs to check for consistency and bound variables at every state: theoretically, since we only require the final plan to be valid, we could simply complete the consistency check on states the planner believes to be goal states. The conventional wisdom has been that checking for inconsistent plans at each state allows pruning of the search space thus expanding fewer nodes to reach the goal; however, this does come at the cost of a higher per-node expansion overhead, especially if the LPs to be solved are large and complex. Our contribution here is propose and evaluate a range of strategies for using schedulers to prune, ranging from the traditional use of a LP for consistency and bounds checking at every state; though to using just an STN to check only tem-

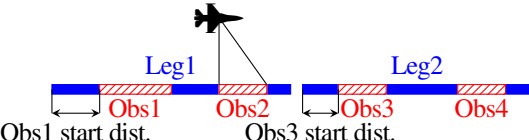

Figure 1: The flying observer

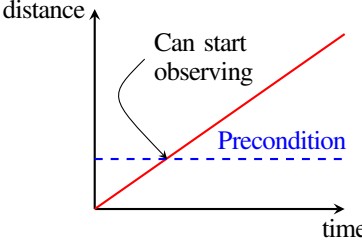

Figure 2: Distance Requirement

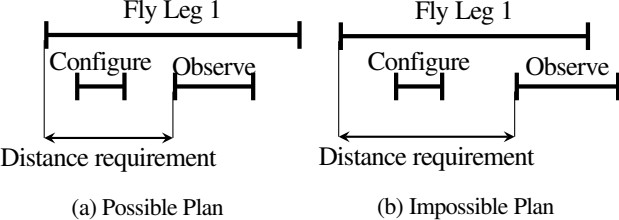

(a) Possible Plan  (b) Impossible Plan

Figure 3: Durative Meaning of Distance Requirement

poral consistency at each state to checking only for consistency in goal states. We thoroughly analyse planner performance with respect to different configurations on our representative application problems and show that we can obtain better performance in these domains with an approach that performs cheaper STN consistency checking on a per-state basis, whilst stil guaranteeing plan consistency using a LP in the goal state.

## 2 Problem Definition

A temporal planning problem with discrete and linear continuous numeric effects is a tuple:

$$\langle I, G, \mathbf{A}, \mathbf{P}, \mathbf{V} \rangle \tag{1}$$

where $\mathbf{P}$ is a set of propositions and $\mathbf{V}$ a set of numeric variables. $I$ is a set of value assignments to these propositions and numeric values, representing the initial state of the problem, $G$ is the goal: a conjunction of propositions in $P$ and linear numeric conditions over the variables in $V$, of the form $w^1 v^1 + w^2 v^2 + ... + w^i v^i \{<, \leq, =, \geq >\} c$ ($w^1 ... w^i$ and $c$ are constants $\in \mathbb{R}$). $\mathbf{A}$ is a set of actions defined by the tuple:

$$\langle d, pre_\vdash, eff_\vdash, pre_\leftrightarrow, eff_\leftrightarrow, pre_\dashv, eff_\dashv \rangle \tag{2}$$

where $d$ is the duration of the action constrained by a conjunction of numeric conditions. $pre_\vdash$ and $pre_\dashv$ are conjunctions of preconditions (facts and numeric conditions) that must be true at the start and end of the action, $pre_\leftrightarrow$ are invariant conditions (preconditions that must hold throughout the action's duration), $eff_\vdash$ and $eff_\dashv$ are instantaneous effects that occur at the start and end of the action. Such effects may add or delete a proposition $p \in \mathbf{P}$ ($eff_+$, $eff_-$) or update a numeric variable $v^i \in V$ according to a linear instantaneous change ($eff_{num}$). In this work all effects in $eff_{num}$ are assumed to be linear and of the form: $v^k \{+=, =, -=\} w^1 v^1 + w^2 v^2 + ... + w^i v^i + w^j * dur_A + c$ where $dur_A$ is a special variable representing the duration of the action $A$ of which this is an effect ($w^1 ... w^j$ and $c \in \mathbb{R}$). $eff_\leftrightarrow$ is a conjunction of continuous effects operating throughout the duration of the action. In this work each continuous effect is linear, i.e. of the form $\frac{dv}{dt} \{+=, =, -=\} c$ where $c \in \mathbb{R}$ is a constant.

## 3 Example Representative Application Domain

This example was supplied by our industrial partner. It encapsulates the structure of problems that arise when using an LP planner in their target application domains.

In this domain, named flying observer, the planner is required to plan a Unmanned Aerial Vehicle (UAV) observation mission. The UAV is required to fly *leg*s over a desired streach of land containing objects to be observed. Each leg is of different length. Each observation has a different duration and requires a different type of equipment. To mark the area within the

leg in which the observation must take place a target-start distance is defined. The observation can take place only when the UAV has flown more than the target-start distance of that leg ($flown_l$ ?$leg \geq$ target-start$_o$). A continous numeric effect of the $fly_l$ action updates the distance flown so far in a leg: $\frac{d flown_l}{dt} = 1$.

Fig 1 illustrates an instance of this domain: in this instance two legs are defined (marked in solid blue lines), in each leg two observations are required (marked in red, pattern filled lines). All observations have a target-start distance defined, but for clarity only the starting distance of the first and third observations are shown.

In order to perform an observation a defined piece of equipment needs to be calibrated and configured for a specific observation. Once the observation is done the equipment needs to be released to become available for future observations. The domain comprises the following actions:

- take-off$_l$: dur = 5; $pre_\vdash$ ={on-ground,first-leg$_l$}; $eff_\vdash$ ={¬on-ground, flown$_l$ = 0}; $eff_\dashv$ ={flying$_l$};
- set-course$_{l1,l2}$: dur = 1; $pre_\vdash$ ={done$_{l1}$,next$_{l1,l2}$}; $eff_\vdash$ ={¬done$_{l1}$}; $eff_\dashv$ ={flying$_{l2}$, flown$_{l2}$ = 0};
- fly$_l$: dur=$^{distance_l}/_{speed_l}$; $pre_\leftrightarrow$ ={flying$_l$}; $pre_\leftrightarrow$ ={flown$_l \leq$ distance$_l$}; $eff_\dashv$ ={done$_l$, ¬flying$_l$}; $eff_\leftrightarrow$ ={$^{dflown_l}/_{dt}$ += 1};
- configure$_{o,e}$: dur = 1; $pre_\vdash$ ={available$_e$, optionfor$_{o,e}$}; $eff_\vdash$ ={¬available$_e$}; $eff_\dashv$ ={configuredfor$_o$, pending$_{o,e}$};
- observe$_{l,o}$: dur = time-for$_o$; $pre_\vdash$ ={configuredfor$_o$, contains$_{l,o}$, awaiting$_o$, target-start$_o \leq$flown$_l$}; $pre_\leftrightarrow$ ={flying$_l$}; $eff_\vdash$ ={¬awaiting$_o$}; $eff_\dashv$={observed$_o$}
- release$_{o,e}$: dur = 1; $pre_\vdash$ ={pending$_{o,e}$}; $eff_\vdash$ ={¬configuredfor$_o$, ¬pending$_{o,e}$}; $eff_\dashv$ ={available$_e$};

The target distance precondition and temporal constraints of this problem force the configure/observe actions to fit within the fly action. The meaning of the precondition is illustrated in Fig 2: The red line is a depiction of the distance change as the UAV flies over the leg. The dashed blue line is the precondition signifying the distance required for the start of the observation.

**Algorithm 1:** Overview of OPTIC's Search

**Data:** Planning Problem $\langle I, G, A, P, V \rangle$
**Result:** A solution plan

1   $Q \leftarrow [I]$;
2   **while** $Q$ *is not empty* **do**
3      $S \leftarrow$ pop the next state from $Q$;
4      $app \leftarrow a \in A_{snap} \cdot S \models pre(A)$;
5      **foreach** $a_i \in app$ **do**
6          $S' = \text{apply}(a, S)$;
7          **if** $\neg$ *isConsistent(S')* **then continue**;
8          ;
9          **if** $S' \models G$ **then return** *plan to* $S'$;
10         ;
11         $UpdateNumericVariableRanges(S')$;
12         $h(S') = ComputeHeuristicValue(S')$;
13         **if** $h(S') \neq \infty$ **then** $Q.enqueue(S')$;
14         ;

15   **return** *problem unsolvable;*

When the distance reaches the value required in the precondition the observation can start. The meaning of the numeric constraint as it is manifested in the temporal state can be seen in Fig 3a.

Notice that by defining contains$_{l,o}$ for multiple legs the planner can be given a choice of multiple legs in which it can decide to perform observation $o$. Further, notice that the legs must be flown in the order defined by the predicates next$_{l1,l2}$, therefore, a leg may not be skipped even if it does not contain an observation.

## 4   Linear Programming (LP) based planners

The mechanism described here allows planners to reason with continuous numeric change. It was first described in COLIN (Coles et al. 2012), then used in POPF and OPTIC. An overview of this search is given in Algorithm 1.

### 4.1   Search In OPTIC

OPTIC performs forward search from the initial state and branching over applicable actions, exploring partially-ordered but un-time-stamped sequences of snap-actions. Snap-actions are instantaneous actions marking the start ($A_\vdash$) and end ($A_\dashv$) of a durative action $A$: $A_\vdash$ has preconditions $pre_\vdash$ A A and effects $eff_\vdash$ A; $A_\dashv$ has preconditions $pre_\dashv$ A and effects $eff_\dashv$ A. For brevity of notation we define the set $A_{snap}$ to contain all snap actions corresponding to the start and end of actions in $A$ ($A_\vdash$, $A_\dashv$ such that $a \in A$) and all instantaneous actions in $A$.

Each state $S$ in search comprises the set of propositions ($S.p \subseteq P$) that are true in $S$ and optimistic upper ($S.max(v)$) and lower ($S.min(v)$) bounds on the value each variable in $V$ can hold in $S$. In the initial state all variables have $max(v) = min(v) =$ the value of $v$ specified in the initial state; $max(v)$ and $min(v)$ will only differ from each other in subsequent states if/when a variable has been subject to continuous change in the plan so far, as the value of $v$ will then change as time elapses in $S$. Search proceeds by popping the first state from the openlist: in our work, we use WA* (W=5) so sort the openlist by $h(S) + 5.g(s)$, using the temporal-numeric RPG heuristic of COLIN (Coles et al. 2012).

At line 4 the planner identifies the actions applicable in $S$, i.e. those whose preconditions are satisfied in $S$ (and do not lead to a state in which the invariants of the currently executing actions are violated). A propositional precondition $p$ is satisfied if it is true in the state, i.e. $p \in S.p$. Numeric preconditions, of the form $w^1 v^1 + w^2 v^2 + ... + w^i v^i \{\geq, >, =, <, \leq\} c$, are deemed satisfied if the values of $max(v)$ and $min(v)$ optimistically satisfy them: that is, for example, we consider $v^1 - v^2 >= 5$ to be satisfied if $ub(v^1) - lb(v^2) \geq 5$. Next (line 6) we use the applicable actions to generate all successors $S'$ of $S$ by adding/deleting all propositions in $eff_a^+$ and $eff_a^-$ respectively, and applying all discrete numeric effects to both $max(v)$ and $min(v)$ for all $v \in V$ affected by $eff_a^{num}$. At this point OPTIC also adds the necessary *ordering* constraints to the plan: the action that has just been applied is ordered after the last actions to add each of its preconditions, after actions whose preconditions it deletes, and after actions with numeric effects on variables it updates or refers to in preconditions/effects: all such constraints are of the form $t_j - t_i \geq \epsilon$, where $t_j / t_i$ are the times at which the new and existing action must occur respectively and $\epsilon$ is a small constant enforcing separation.

### 4.2   Checking Temporal/Numeric Consistency

Whilst the plan generated by the above search is guaranteed to be *propositionally consistent* (all propositional preconditions are satisfied when an action is applied) it might not be *temporally or numerically consistent*. Temporal/numeric consistency of plans must be explicitly checked, this is done at Line 7, using either an STN or LP as appropriate. To illustrate why this in the case, consier the partial plan: take-off,fly$_{l0}^\vdash$, configure$_{o1,e2}^\vdash$, configure$_{o1,e2}^\dashv$,fly$_{l0}^\dashv$ in our application domain. This plan is propositionally sound, but if the duration of configure exceeded the duration of fly then this would not be a *temporally consistent* plan.

First, consider the case where there are no continuous or duration dependedn numeric effects in the plan (like the one given above). In this case we can update the values of all numeric variables in each state according to discrete numeric effects, and know the exact values of $v_i$ in each state, allowing us to enforce numeric preconditions correctly during search ($max(v) = min(v) =$ the known value of v in S). The only remaining constraints we have are temporal constraints: ordering constraints of the form $t_j - t_i \geq \epsilon$ and duration constraints of the form $t_{a\dashv} - t_{a\vdash} \{\leq, =\geq\} c$, where c is a constant (without loss of generality, c can be computed from the known values of variables $v^i \in v$ in the state in which $a_\vdash$ was applied). It is well known that such a set of constraints constitutes a simple temporal network (STN) (Dechter, Meiri, and Pearl 1991): a solution to this represents a valid schedule for the snap-actions in the plan. This can be solved (or proven unsolvable) in polynomial time using an all-pairs shortest path algorithm.

Consider now the more complex case where we have a partial plan that has a precondition on a variable that has undergone continuous numeric change, e.g.: take-off,fly$_{l0}^\vdash$, configure$_{o1,e2}^\vdash$, configure$_{o1,e2}^\dashv$, observe$_{l0,o1}^\vdash$, observe$_{l0,o1}^\dashv$,fly$_{l0}^\dashv$. We can as above, equally check that the plan is temporally consistent (the duration of configure and observe actions are less than the fly action) using a simple temporal network; if the plan were temporally inconsistent we could prune it using only an STN. However, we cannot be certain using an STN that this plan is valid as the

STN does not take into account the precondition of observe$_{l0,o1}$: target-start$_{o1} \leq$ flown$_l$. It might be that there is insufficient time within the leg $l0$ action to wait until this constraint is met (e.g. if leg $l0$ has duration 15 and observe$_{l0,o1}$ duration 10 and target-start distance 11) then the plan would not be temporally and *numerically* sound (this is illustrated in Fig 2–3). In this case we need to use an LP to encode both the temporal and numeric constraints of the problem. Section 5 details how this LP is built in such a way that a solution is a valid assignment of time stamps to the snap actions in the partial plan that satisfies the temporal and numeric constraints of the problem.

At line 7 OPTIC intelligently selects the scheduler used based on the constraints in the plan and will use the cheaper STN by default, using the LP only when constraints that necessitate it are present. If the LP, or STN, determines that there is no consistent schedule for the plan reaching a state, the plan is temporally or numerically invalid and so the state is pruned. If the LP determines there is a valid schedule of the actions that satisfies the temporal/numeric constraints then $S'$ is a valid successor of $S$. If $S'$ satisfies the goal, we have as solution plan. Otherwise, the LP is used again to find the range on each state variable (line 11) prior to heuristic evaluation, using the standard temporal/numeric relaxed planning graph heuristic of Colin (Coles et al. 2012). It is then inserted into the openlist, providing $h(S') \neq \infty$, i.e. the heuristic does not indicate $S'$ is a dead-end.

To understand this use of the LP consider the partial plan: take-off,fly$_{l0}^{\vdash}$, fly$l0^{\vdash}$, configure$_{o1,e1}^{\vdash}$, configure$_{o1,e1}^{\dashv}$. As soon as the action fly$_{l0}^{\vdash}$ is applied we have a continuous effect increasing the value of the numeric variable flown$_{l0}$, so we can now no longer say flown is in the range [0,0] (as in the initial state) but it could be anywhere in the range $[0,\infty]$ as an effect increasing it has started but not yet ended. In the general case we can exploit the same LP we used for consistency checking to tighten the bounds on numeric variables in each state: for each $v \in V$ that has been subject to any continuous or duration-dependent change in the plan to reach $S'$ we set the LP objective function to maximise (and then minimise) $v$ to find out the maximum possible value of $v$ admitted by the plan. Again, we explain how the LP is used to do this in Section 5.

We make two observations about this use of the LP to tighten variable bounds. First it is effectively optional: even if we assumed the bounds on $v$ were $[0,\infty]$ then when $S'$ is later expanded, and isConsistent() is used on its successors, this would prune any reached by an action whose preconditions were not satisfiable in $S'$. Though, the use of the LP to generate tighter bounds on the value of $v$ helps prune the list of applicable actions: if an action can be pruned because its numeric preconditions are unsatisfied according to the tighter bounds on $v$, this saves building another LP and using isConsistent for the corresponding successor: this is a key trade-off that we investigate in this paper. The second observation is that these bounds are optimistic: that is, we ask the LP to maximise $v$ and separately to minimise $w$ however, it might be the case that the maximum value of $v$ and the minimum value of $w$ are not achievable by the same schedule; so the precondition $v - w > = 5$ might still not be satisfiable even if the bounds state that it is; thus we always need the is consistent check upon expanding $S'$ to confirm its preconditions are satisfied; although this would be needed anyway, to ensure any other tem-

| Step | Action | variables | constraints | comment |
|---|---|---|---|---|
| 0 | TakeOff | $t_0$ | $\geq 0$ | |
| 1 | Fly$_{l0}^{\vdash}$ | $t_1$ | $-t_0 \geq \epsilon$ | Step1 afer Step0 |
| | | $flown\_l0_1$ | $=0$ | Initial Assignlemt |
| | | $flown\_l0'_1$ | $flown\_l0_1$ | Value after action |
| | | | $\leq distance\_l0$ | Invariant |
| 2 | Configure$_{o1,e1}^{\vdash}$ | $t_2$ | $-t_1 \geq \epsilon$ | Step2 after Step1 |
| | | $flown\_l0_2$ | $=flown\_l0'_1+1*(t_2-t_1)$ | Value before action |
| | | | $\leq distance\_l0$ | Invariant |
| | | $flown\_l0'_2$ | $=flown\_l0_2$ | Value before Action |
| | | | $\leq distance\_l0$ | Invariant |
| 3 | Configure$_{o1,e1}^{\dashv}$ | $t_3$ | $-t_2 \geq \epsilon$ | Step2 after Step1 |
| | | $flown\_l0_3$ | $=flown\_l0'_2+1*(t_3-t_2)$ | Value before action |
| | | | $\leq distance\_l0$ | Invariant |
| | | $flown\_l0'_3$ | $=flown\_l0_2$ | Value before Action |
| | | | $\leq distance\_l0$ | Invariant |
| 4 | Observe$_{o1,l0}^{\vdash}$ | $t_4$ | $-t_3 \geq \epsilon$ | Step3 after Step2 |
| | | $flown\_l0_4$ | $=flown\_l0'_3+1*(t_4-t_3)$ | Value before action |
| | | | $\geq target\text{-}start\_o1$ | Start precondition |
| | | | $\leq l0\_dist$ | Invariant |
| | | $flown\_l0'_4$ | $=flown\_l0_4$ | Value after action |
| | | | $\geq Target\_dist\_o1$ | Strat precondition |
| | | | $\leq l0\_length$ | Invariant |
| 5 | Observe$_{o1,l0}^{\dashv}$ | $t_5$ | $-t_4 \geq \epsilon$ | Step4 after Step3 |
| | | | $-t_4 \leq time-for$ | Action duration |
| | | $flown\_l0_5$ | $=flown\_l0'_4+1*(t_5-t_4)$ | Value before action |
| | | | $\leq distance\_l0$ | Invariant |
| | | $flown\_l0'_5$ | $=flown\_l0_5$ | Value after action |
| | | | $\leq l0\_length$ | Invariant |
| 6 | now | $t_{now}$ | $-t_5 \geq \epsilon, -t_4 \geq \epsilon, -t_3 \geq \epsilon,$ $-t_2 \geq \epsilon, -t_1 \geq \epsilon, -t_0 \geq \epsilon$ | After All Steps |
| | | $flown\_l0_{now}$ | $=flown\_l0'_5+1*(t_{now}-t_5)$ | Value Now |

Table 1: LP Equations of a Partial Plan

poral/numeric constraints are satisfied when applying the action.

## 5   Building LPs in OPTIC

In this section we detail how the LP to check plan consistency is formulated at each state. We refer throughout to Table 1 which shows an example LP for the partial-plan: take-off, fly$_{l0}^{\vdash}$, configure$_{o1,e1}^{\vdash}$, configure$_{o1,e1}^{\dashv}$, observe$_{o1,l0}^{\vdash}$, observe$_{o1,l0}^{\dashv}$.

To represent the planning problem as an LP the following LP variables are defined for each step $i$ of the partial plan: $t_i$ defining the time at which the step is to be taken, $v_i$ the value of variable $v \in \mathbf{V}$ just before the application of the action and $v'_i \in \mathbf{V}$ the value of variable $v$ just after the application of the action.

**Temporal Constraints:** Ordering constraints can be enforced exactly as written, when step $j$ must occur after step $i$ we write:

$$t_j - t_i \geq \epsilon \qquad (3)$$

Duration constraints can also be formulated directly:

$$t_j - t_i \{\geq, \leq, =\} w_i^1 v_i^1 + w_i^2 v_i^2 + ... c \qquad (4)$$

For example, in Table 1 ordering constraints enforce that observe$_{o1,l0}^{\vdash}$ occurs at least $\epsilon$ after configure$_{o1,e1}^{\dashv}$ as the latter achieves a precondition of the former; and duration constraints enforce that observe$_{o1,l0}^{\dashv}$ occurs exactly the defined duration after observe$_{o1,l0}^{\vdash}$ ($t_{Obs1\dashv} - t_{Obs1\vdash} = dur_{Obs1}$). Recall that in the absence of continuous numeric effects and duration-dependent effects, the LP need only contain these temporal constraints, and the right hand sides of all duration constraints are known constants, therefore an STN solver suffices.

**Numeric Constraints:** To manage linear continuous change an additional variable $\delta v_i$ stores the sum of change acting on

variable $v$ after step $i$. If $A$ is a durative action with a continuous linear effect, and $c_A$ the constant defining said linear change (i.e. $\frac{dv_i}{dt} += c_A$). Then $\delta v_i$ is calculated thus:

$$\delta v_i = \begin{cases} \delta v_{i-1} + c_A & \text{if } A_i = A_\vdash \\ \delta v_{i-1} - c_A & \text{if } A_i = A_\dashv \end{cases} \quad (5)$$

This, if $A$ starts at $step_i$, $c_A$ will be added to $\delta v_i$. If $A$ ends at $step_j$, $c_A$ will be removed from $\delta v_j$. Thus $\delta v_i$ is the sum of all effects currently active on $v \in \mathbf{V}$ after $step_i$. We use $\delta v_i$ to compute the value of variables undergoing continuous numeric change:

$$v_i = v'_{i-1} + \delta v_{i-1}.(t_i - t_{i-1})$$

This can be seen in Table 1: the only continuous effect is acting on $flown\_l0_i$ from step 2 onward. Thus $\delta flown\_l0_i = 1$ (for $i \geq 2$) and the value of $flown\_l0_i$ is calculated according to this at each step (e.g. $flown\_l0_2 = flown\_l0'_1 + 1.(t_2 - t_1)$).

Numeric preconditions $pre_\vdash$ and $pre_\dashv$ are formulated over the respective variables $v_i$. We formulate the invariant conditions $pre_\leftrightarrow$ of actions at the start and end steps of the action and at every step between them. This is sound because all effects are linear so no turning points can be present between the steps.

In the example LP for the flying observer (Table 1) the distance-flown ($flown_{l1}$) precondition on the $observe^\vdash_{o1,l1}$ action is encoded at step3 over the variable ($flown\_l0_3$); and the invariant ($flown_{leg_j} \leq distance_{l0}$) of $fly_{l0}$ can be seen enforced immediately after $step_1$ (i.e. on $flown\_l0'_1$) and before and after all subsequent steps that refer to $flown\_l0$ (over $flown\_l0_i$ and $flown\_l0'_i$).

A solution to this LP gives us an assignment to each $t_i$ representing a valid time stamp for each $step_i$ in the partial plan; if the LP solver reports that no solution exists then the plan is inconsistent and we can prune the resulting state. If the state is deemed consistent the formulated LP problem is used again (Algorithm 1, Line 11) to determine bounds on the numeric variables. To do this we create new step now, with associated timestamp variable $t_{now}$ ordered after all existing steps; then for each variable $v \in \mathbf{V}$ we create $v_{now}$, and calculate its value at $t_{now}$ in the usual way. The LP is solved with an objective minimise (then again to maximise) $v_{now}$ yielding the minimal (maximal) possible value $v$ may hold in the current state. Table 1 illustrates this for the variable $flown\_l0$. $t_{now}$ is constrained to come after all other plan steps, and the value of $flown\_l0_{now}$ is computed as $flown\_l0'_5 + 1.(t_{now} - t_5)$; using the objective minimise (maximise) $flown\_l0_{now}$ will tell us the maximum and minimum feasible values of $flown\_l0$ we can expect to rely on for the precondition of any action to be applied in this state.

Prior work observed that the formulation of the LP was expensive and solving was cheap (Coles et al. 2012). Therefore since the LP is not reformulated, but simply resolved for different $v_{now}$s, computing bounds was relatively inexpensive. However, this did not match our observations when examining our complex real-life problems with large LPs[1].

## 6 LP vs STN Scheduling in OPTIC

It is well known that the scalability of planners is affected by both the number of applicable actions per state (branching

---

[1]OPTIC's LP building code is also more optimized than COLIN's

---

| Instance | Observations | Legs | Observations Required in Goal |
|---|---|---|---|
| 1 | 10 | 28 | 4 |
| 2 | 15 | 38 | 6 |
| 3 | 20 | 48 | 8 |
| 4 | 25 | 58 | 10 |
| 5 | 30 | 68 | 12 |

Table 2: Single Observation Per-Leg Instances

factor) and the length of the required solution plan (depth to which the search tree must be explored). The latter of these factors is magnified in planners using LP schedulers because the size of the LP being solved increases with the length of the plan being scheduled. In this section we explore the scalability of LP based planners in our complex real-life observer domain, which involves continuous numeric change and requires long solution plans compared to conventional benchmark domains.

### 6.1 Standard OPTIC Performance

To explore this behaviour we ran the flying observer domain on 5 instances increasing in difficulty. In this domain, there is never more than one observation that can be chosen to happen in each leg (i.e. if contains$_{l1,o1}$ is defined then there does not exist any other observation o$_i$ such that contains$_{l1,oi}$ is defined). Further, there is no choice over which leg each observation can occur in (i.e. if contains$_{l1,o1}$ is defined then there does not exist any other leg $l_i$ such that contains$_{li,o1}$ is defined). Each observation requires one piece of equipment out of the three available. The number of observations defined, the number of legs, and the number of observations required in the goal differs between the instances as specified in Table 2.

Note that to emulate a real life scenario where the instance of the problem may contain many optional actions, not all of which are required for the goal, the number of observations and legs here is much greater than is required for the goal. For instance, leg number 25 and higher is not required, yet is a possibility that the planner might consider. The same goes for observation number 20 - it is defined, but not required for the goal. Solving the hardest instance of this domain was slow and took OPTIC about 260 seconds on an Intel i7 2.80GHz.

A second variant of the domain, again representing an additional challenge encountered in our application, adds a global variable counting the total distance flown, and a precondition to the "configure" action requiring that this is not greater than 1000 units. The changed actions would therefore be:

1. fly$_l$: dur=$^{distance_l}/_{speed_l}$; $pre_\vdash$ ={flying$_l$}; $pre_\leftrightarrow$ ={flown$_l \leq$ distance$_l$}; $eff_\dashv$ ={done$_l$, ¬flying$_l$}; $eff_\leftrightarrow$ ={$^{dflown_l}/_{dt} += 1$, $^{dtotalFlown}/_{dt} += 1$};

2. configure$_{o,e}$: dur = 1; $pre_\vdash$ ={available$_e$, optionfor$_{o,e}$, totalFlown$\leq$1000}; $eff_\vdash$ ={¬available$_e$}; $eff_\dashv$ ={configuredfor$_o$, pending$_{o,e}$};

Before adding this variable the "configure" action was entirely propositional, and therefore consistency of plans containing it could be confirmed using an STN. Now it has a precondition that inspects a variable affected by continuous numeric change so, as discussed in Section 4, any plan containing this action requires a LP to confirm consistency.

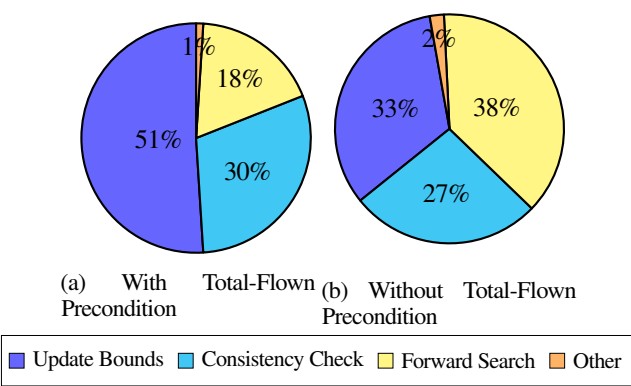

(a) With Total-Flown Precondition (b) Without Total-Flown Precondition

■ Update Bounds ■ Consistency Check ■ Forward Search ■ Other

Figure 4: Single Observation Per Leg Profiling (Instance 5)

| | # | Nodes Generated | Nodes Expanded | Nodes Evaluated | Nodes STN Checked | Nodes STN Pruned | Nodes LP Checked | Nodes LP Pruned |
|---|---|---|---|---|---|---|---|---|
| Normal | 1 | 413 | 41 | 407 | 309 | 0 | 309 | 2 |
| | 2 | 4976 | 395 | 4750 | 4855 | 0 | 4855 | 222 |
| | 3 | 28465 | 1570 | 28032 | 28315 | 0 | 28315 | 429 |
| | 4 | 92358 | 6683 | 91115 | 92195 | 0 | 92195 | 1239 |
| | 5 | 135043 | 10479 | 133251 | 134879 | 0 | 134879 | 1788 |
| LP for Goals | 1 | 413 | 41 | 409 | 409 | 0 | | |
| | 2 | 4976 | 395 | 4972 | 4972 | 0 | | |
| | 3 | 28465 | 1570 | 28461 | 28461 | 0 | | |
| | 4 | 92358 | 6683 | 92354 | 92354 | 0 | | |
| | 5 | 135043 | 10479 | 135039 | 135039 | 0 | | |

Table 3: Nodes in Search, Single Observation Per Leg

Solving the hardest instance without the total-flown variable (the initially described domain) took about 260 seconds 60% of which were spent solving the LP. However, when solving the hardest instance with the total-flown variable and precondition, planning took about 560 seconds 81% of which were spent in the LP. Fig 4 shows the profiling results of the 5th instance in the domain with the total-flown precondition and without. STN solving takes negligible time, and is included in the 'Forward Search' measurement; LP takes the most time, particularly with 'total-flown'. Inspecting further, the more nodes generated during search (the more LPs solved) the slower it is. For instance, in general it was seen that running the domain lacking the total-flown requirement was faster, however, in instance 3 of the domain, more nodes were visited during search, and the runtime on that instance specifically was slower.

Also notable is that in our domain, more time is spent computing variable bounds than consistency checking. This suggests these LP calls are expensive, so avoiding them may be beneficial.

## 6.2 Proposals to Improve Performance

Since the LP solving is computationally expensive, we try to reduce the number of times the LP solver is called during search. Four options for reducing the amount of calls for the solver are presented here:

1. Call updateVariableRanges (Algorithm 1 Line 11) only for the variables which have an active continuous effect acting on them in the partial plan (i.e. an action with a continuous effect on $v$ has started, but not yet finished);

2. Solve the LP only to check for consistency (Line 7), and don't call updateVariableRanges for any variables (i.e. skip Line 11);

3. Solve an STN to check only *temporal consistency* (at Line 7) even if an LP would normally be used. Solve the LP to check for consistency only in possible goal states (i.e. at Line 9);

4. Solve neither the LP nor an STN at each state (remove Line 7), and solve the LP only in possible goal states.

The first option, updating only the active variables, reduces the number of LPs solved in each state by only increasing bounds due to effects, but not always tightening them based on the preconditions. A continuous change can increase the bounds on a variable, whereas preconditions may only tighten the

bounds. In the default configuration OPTIC updates bounds on all variables that have ever been subject to continuous numeric change; here we suggest optimizing only the variables that are acted upon by currently executing actions. This is a compromise: by checking the effect we guarantee the most optimistic bounds, while reducing the number of LPs solved.

The second option is to solve the LP only for consistency checks (i.e. do not use the LP to update variable bounds). This means fewer actions will be marked as inapplicable, and the pruning of branches is done based on consistency only. Updating bounds is done to narrow the search space; however, if the cost of this is more expensive than the search effort saved this is not worthwhile.

The third option is to solve the LP only on the states which the forward search finds as possible goal states. This means that fewer actions are marked as inapplicable during the search and the states are only tested for temporal consistency, not for numeric consistency, until branching reaches a goal state. Search finds an ordering of actions that transform the initial into a goal state, here the LP solver is called to schedule the ordered actions. If a schedule is found then this indeed is a goal state. If a schedule cannot be found than this is an invalid branch and the planner needs to backtrack.

The fourth option is ploughing through forwards search disregarding any temporal or numerical constraints, then, when a possible goal state is reached, solve a LP to check whether it can be scheduled. When this method finds a solution, it is not much faster than method 3. Sadly though, it almost never finds one. In the domains we checked this method was only able to solve the first and second instances of the single observation-per-leg domain, and did so quite quickly. However, it timed out on all other instances and domains we checked. Because hardly any pruning is done it is quite easy for search to get stuck in a fruitless subspace when this method is employed. It is therefore not recommended, and is not shown in the results.

## 6.3 Comparison of Proposals

Fig 5 presents the results of the first three options on the 5 instances. Fig 5a lists the results on the domain with the total-flown precondition on the "configure" action, and Fig 5b without.

The slowest method, named "Normal" (blue, northeast to southwest pattern) is running OPTIC with the default configuration: solving the LP for consistency, then several more times to find the bounds on the numeric variables, and is always the most computationaly expensive. Next (olive, dotted pattern) is the

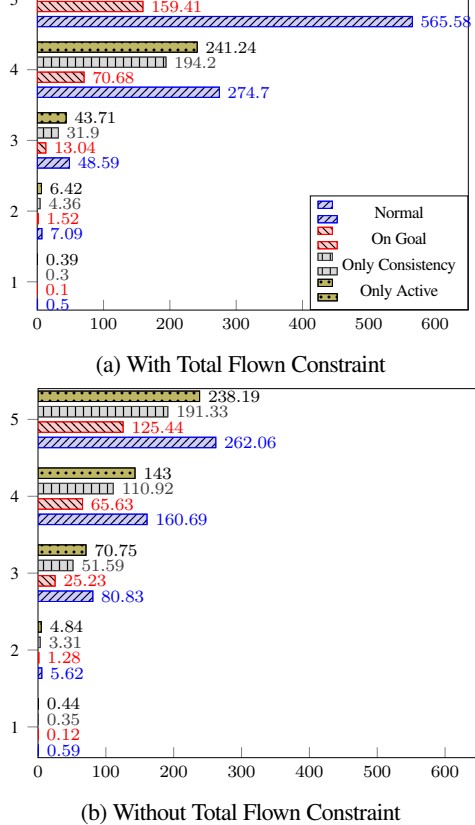

(a) With Total Flown Constraint

(b) Without Total Flown Constraint

Figure 5: Single Observation Per Leg Runtime (seconds)

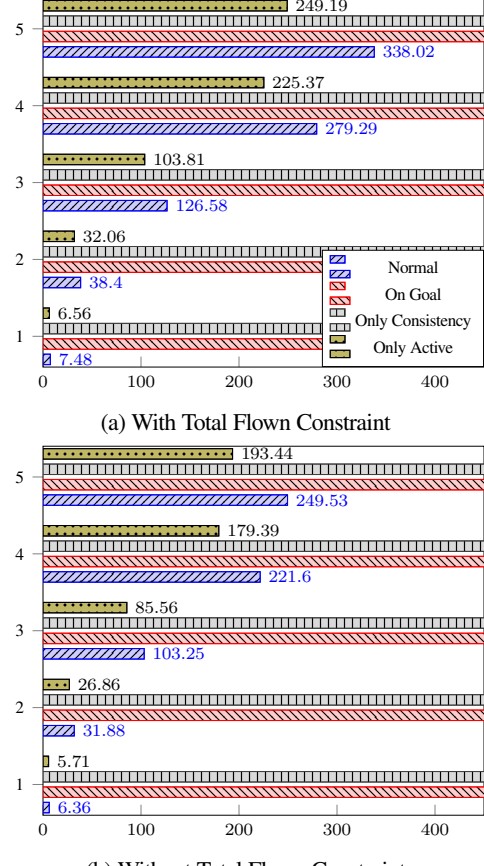

(a) With Total Flown Constraint

(b) Without Total Flown Constraint

Figure 6: Multiple Observation Per Leg Runtime (seconds)

option that solves the LP for consistency checking, but updates only the bounds of the variables currently subject to continuous numeric change. Faster still is the option that solves the LP only for consistency checking (gray, horizontal pattern) but does not update bounds. Finally, the fastest option on these instances is the third option: solving the LP only at at the goal state while using STN in non-goal states for the consistency check (marked red, with northwest to southeast pattern). As stated before the 4th option performance was too poor to include here.

Table 3 shows the number of nodes generated, expanded and evaluated during the search, as well as the number of times STN and LP solvers were called for consistency checks and the number of times each found a branch to be inconsistent. The number of nodes examined in the default configuration was identical to those in the update of the active variables only and when no update took place, and therefore are not shown here. The fact the number of nodes were similar suggests that setting and updating the bounds on the variables did not help flagging actions as invalid, and therefore, in these domains and problem instances slowed the planning process down. In addition, the number of nodes examined when calling the LP only on the goal state was not greatly different to the normal run, suggesting the consistency check of the LP also did little to prune branches.

These results show that in this case, whilst the small cost of the STN is outweighed by significant search pruning; it is most beneficial to avoid LPs as much as possible using them only when necessary, to check whether a goal is valid or not. However, not solving the LP might have a hindering effect on the search.

Fig 7 supports the above claim. It presents the profiling of the different methods. It shows the correlation between the total runtime and the call for the LP solver. The less time is invested in the solving of the LP the faster the planner reaches the goal state.

The above claim, however, cannot be said to be generally true. We tested the same two domains with a different set of problem instances which also represent a problem that may arise in real life. Here, instance n-1 contains n legs (0..n-1) with 6 observations that must take place in each leg (i.e. for $k \in 1...n$, $i \in [6k, 6k+5]$ contains$_{lk,oi}$) all of which are required in the goal (and require a different one of 6 pieces of equipment). There is no choice over which leg each observation can occur in (i.e. if contains$_{l1,o1}$ is defined then there does not exist any other leg $l_i$ such that contains$_{li,o1}$ is defined); except for $o_5$, which can be performed in either leg$_0$, or leg$_{n-1}$ (contains$_{l1,o5}$ and contains$_{ln-1,o5}$). To compensate for this $leg_{n-1}$ has only 5 additional observations ($o_{6n}$ to $o_{6n+4}$) rather than the usual 6. The instances are engineered such that the duration of leg$_0$ is too short for $o_6$ to fit inside: target-start$_{o6}$ is defined such that waiting long enough for (target-start$_{o6} \leq$ flown$_{leg1}$) to be satisfied, means $o_6$ cannot finish within $leg_0$ as illlustrated in

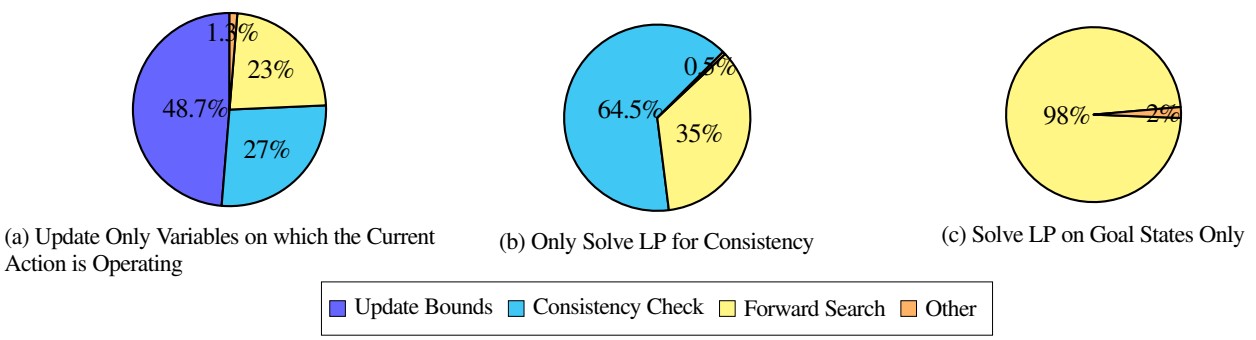

(a) Update Only Variables on which the Current Action is Operating

(b) Only Solve LP for Consistency

(c) Solve LP on Goal States Only

■ Update Bounds  ■ Consistency Check  □ Forward Search  ■ Other

Figure 7: Profiling Suggested Improvements, Single Observation Per-Leg, Instance 5

| Instance | Nodes | | | Nodes STN | | Nodes LP | |
|---|---|---|---|---|---|---|---|
| | Generated | Expanded | Evaluated | Checked | Pruned | Checked | Pruned |
| 1 | 3510 | 1316 | 3393 | 3470 | 112 | 3358 | 0 |
| 2 | 11170 | 3631 | 10956 | 11111 | 208 | 10903 | 0 |
| 3 | 24639 | 7778 | 24280 | 24523 | 352 | 24171 | 0 |
| 4 | 38244 | 11309 | 37812 | 38108 | 425 | 37683 | 0 |
| 5 | 38981 | 11336 | 38549 | 38840 | 425 | 38415 | 0 |

Table 4: Nodes in Search, Multi Observation Per Leg

Fig 3b. $leg_n$ is, however, long enough to accomodate $o_6$ with the precondition satisfied.

Fig 6 presents the results of these multiple-observations per leg instances. Fig 6a on the domain with the precondition for the total-flown in the "configure" action, and Fig 6b without. Table 4 presents the number of nodes using option 1 and the normal configuration, as did Table 3 in the previous set of instances.

As can be seen in Fig 6, on these instances when solving without checking for consistency at each state (option 3) the planner timed out (1000 seconds). Inspection of the states the planner visited during the forward search revealed that this is because the planner added the invalid action to the first leg, and went on searching down the branch. When it reached the goal state it found that it was inconsistent, and started backtracking, but never backtracked enough to find the valid solution. This can also be inferred from Table 4, unlike in the previous case, search makes use of the updated bounds on the variables to prune some branches, similarly actions are being marked as innaplicable in the search (specifically it can infer $target - start_{o6} \leq flown_{leg1}$ will never be satisfied so does not consider putting $o_6$ in $leg_0$).

When solving using the second option, the planner timed out as well. This is because the planner kept trying to add the non-valid action, and found it inconsistent. Since there are ordering constraints on the observations, the planner would try to fit the inconsistent observation as the first observation, then, when failed, it would try it as the second, then the third, and so on. This is an expensive process, and it timed out as well.

In this instance updating the bounds was beneficial. Updating only for the active variables was more efficient, as this guaranteed that each variable would be updated at least once, and therefore narrowed down the search space by marking the problematic observation as a non valid action for the first leg.

## 7 Discussion and Conclusions

The profiling results presented here suggest that the Simple Temporal Network (STN) is much faster than the LP solver. In the domain discussed, which was supplied by an industrial partner, it was shown that the time to solve many LPs which grow in size may not be negligible, and may lead to the planner having difficulties reaching a solution.

Four options were proposed for reducing planning time: updating the bounds on fewer variables by selecting only those currently undergoing continuous numeric change; not updating the bounds at all (solving the LP only for consistency); using an STN on non goal states (while solving the LP only on possible goal states) and not using any solver on a non-goal state (calling the STN and LP only at a goal state).

Solving the STN on non-goal states allows a large number of states to be explored quickly. However, this is only useful in the cases in which actions cannot be marked as inapplicable by their numerical preconditions (they can only be marked as such by their propositions). Using this option in problems with non-cosmetic numeric precondition might cause the planner to search down a branch that is not valid, and to remain in that branch for too long to practically be able to reach a solution.

Solving the LP only to check consistency speeds search up by avoiding LP calls to determine variable bounds. But, again, this increases the branching factor, generating more states, thus slowing the search down.

The last option, not calling any solver on non-goal, was shown to be inefficient, it carries no advatages over the others.

Finally, updating only the active variables was found to be a good compromise. It reduces the per-state LP overheads compared to the default configuration of OPTIC, with a net reduction in planning time; in principle, it has a higher branching factor, so it is not guaranteed to pay off, but we did not encounter such a case in this work.

These four options have been implemented in an updated version of OPTIC, allowing the user to choose from them if needs be. Future research would involve the automatic identification of cases in which per-node LP solving is non beneficial, and the selective update of the variables to facilitate faster search.

## Acknowledgements

This work was supported by the UK Engineering and Physical Sciences Research Council (EPSRC) grant EP/R511559/1

(Deployment of Expressive Continuous Numeric Planners in Large Scale Applications).

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
