# OpenReview forum: "Evaluating the Cost of Employing LPs and STPs in Planning: Lessons Learned From Large Real-Life Domains"
_icaps-conference.org/ICAPS/2019/Workshop/SPARK — SPARK 2019_

### Official Review · AnonReviewer2 · 2019-05-01
**Good contribution. Deeper analysis correlating problem characteristics to performance would help generalize to other domains.**

**Rating:** 4
**Confidence:** 2

**Review:**

When building a planner, or any software, tradeoff between different options need to be evaluated and systematic comparison of the 4 additional options on how frequently LP model is setup and call during the forward search are valuable lesson for any users of OPTIC who plan to use it for solving domains with extensive set of temporal and numerical constraints. The fact that those options are now build into OPTIC and are available for other users is also a plus.

The analysis is only carried out on a single domain and only compared the performance of different options. I would like to see a deeper analysis to identify and correlate particular characteristics of this domain with the observed performance of different options. Then users working on other domains can see how the results in this paper can generalize to their domains. The authors mention as their future work techniques to automatically identify the best option for a given domain, I guess through automatic identification of certain domain characteristics; it would be nice to elaborate more on how the authors plan to achieve that in the final version of the paper.

Extend the evaluation to a more diverse set of benchmarks with different characteristic also will be valuable. I hope that is one of the future work too.

Minor comments:
-	I feel “UAV-based Surveilance Domain” instead of “Large Real-Life Domains” better reflects the experiments carried out in this paper.
-	STN is much faster than LP is not a surprise. It’s quite simple to implement STN consistency checking routine while LP solver is very complicate and for the LP part I assume that OPTIC needs to call an external solver through its API.

---

### Official Review · AnonReviewer1 · 2019-05-02
**STP checking works well in many real application of automated planning.**

**Rating:** 3
**Confidence:** 2

**Review:**

This paper considers real-life planning problems characterized by temporal/numeric constraints and analyses its resolution process via a forward-search planning strategy based on a step-by-step application of a Linear-Programming (LP) procedure for consistency checking.   On one hand, the LP checking performed during the search process allows the pruning of many infeasible branches; on the other, it does carry high computational costs. In this paper authors investigate the possibility for the planners to call the LP solver less often and using, at each search step, a cheaper consistency check based only on the STP (Simple Temporal Problem) constraints.

The paper is well written and well organized and proposes a relevant topic to the workshop. Despite the proposed results seems relevant for the specific area of application considered in the work, given the use of an STP solver as a basic mechanism for  consistency checking is common in several timeline-based planners, it could generate a limited interest in the workshop participants.

---

### Decision · Program_Chairs · 2019-05-08
**Acceptance Decision**

**Decision:**

Accept

**Comment:**

Beneficial to see how algorithms work in real-life domains.